# Evaluating the State of Glomerular Disease by Analyzing Urinary Sediments: mRNA Levels and Immunofluorescence Staining for Various Markers

**DOI:** 10.3390/ijms25020744

**Published:** 2024-01-06

**Authors:** Kojiro Yamamoto, Takashi Oda, Takahiro Uchida, Hanako Takechi, Naoki Oshima, Hiroo Kumagai

**Affiliations:** 1Department of Nephrology and Endocrinology, National Defense Medical College, Tokorozawa 359-8513, Japan; koujinnnn@yahoo.co.jp (K.Y.); hanako-t@mte.biglobe.ne.jp (H.T.); oshima@ndmc.ac.jp (N.O.); hiroo-kumagai@saitama-sekishinkai.org (H.K.); 2Department of Nephrology, Tokyo Medical University Hachioji Medical Center, Hachioji 193-0998, Japan; tu05090224@gmail.com; 3Department of Nephrology, Sayama General Clinic, Sayama 350-1305, Japan

**Keywords:** claudin1, macrophage, M2 macrophage, parietal epithelial cell, urinary sediment

## Abstract

Renal biopsy is the gold standard for making the final diagnosis and for predicting the progression of renal disease, but monitoring disease status by performing biopsies repeatedly is impossible because it is an invasive procedure. Urine tests are non-invasive and may reflect the general condition of the whole kidney better than renal biopsy results. We therefore investigated the diagnostic value of extensive urinary sediment analysis by immunofluorescence staining for markers expressed on kidney-derived cells (cytokeratin: marker for tubular epithelial cells, synaptopodin: marker for podocytes, claudin1: marker for parietal epithelial cells, CD68: marker for macrophages (MΦ), neutrophil elastase: marker for neutrophils). We further examined the expression levels of the mRNAs for these markers by real-time reverse transcription polymerase chain reaction. We also examined the levels of mRNAs associated with the M1 (iNOS, IL-6) and M2 (CD163, CD204, CD206, IL-10) MΦ phenotypes. Evaluated markers were compared with clinical and histological findings for the assessment of renal diseases. Claudin1- and CD68-positive cell counts in urinary sediments were higher in patients with glomerular crescents (especially cellular crescents) than in patients without crescents. The relative levels of mRNA for CD68 and the M2 MΦ markers (CD163, CD204, CD206, and IL-10) in urinary sediments were also higher in patients with glomerular crescents. These data suggest that immunofluorescence staining for claudin1 and CD68 in urinary sediments and the relative levels of mRNA for CD68 and M2 MΦ markers in urinary sediments are useful for evaluating the state of glomerular diseases.

## 1. Introduction

The number of patients with renal disease who progress to end-stage renal failure is increasing on an annual basis. This results in increased medical costs. Predicting the progression of renal disease and estimating disease activity are important in the treatment of renal diseases. The degree of kidney injury can be assessed by clinical indicators or histological evaluation, and the results of these evaluations are not always in agreement. This is because clinical indicators are generally functional assessments, such as serum creatinine levels, and may reflect acute, temporary changes, such as hemodynamic changes. On the other hand, histological evaluation by renal biopsy is the gold standard for making the final diagnosis and for predicting the progression of renal disease. Especially, the presence of glomerular crescents in renal biopsy specimens is one of the most important factors for predicting the prognosis of patients with renal disease [1,2,3], but monitoring disease status by performing biopsies repeatedly is impossible because a renal biopsy is an invasive procedure that carries considerable risk. Moreover, focal lesions such as crescents or segmental sclerosis may be overlooked in the limited biopsy specimens [4]. 

A urine test is non-invasive and may reflect the general condition of the entire kidney better than the results of a renal biopsy [5]. Urinary sediments may contain tubular epithelial cells, glomerular epithelial cells (parietal epithelial cells (PECs) and podocytes), and inflammatory cells, and although several studies have examined podocytes or inflammatory cells in urinary sediments, few have extensively examined the renal parenchymal cells in urinary sediments [6,7,8,9,10,11,12,13]. We therefore investigated the diagnostic value of extensive urinary sediment analysis by immunofluorescence (IF) staining for markers expressed on kidney-derived cells. We used cytokeratin as a marker for tubular epithelial cells and synaptopodin as a marker for podocytes. And since PECs have been reported to specifically express the tight-junction protein claudin1 [14], we used it as a marker for PECs. We hypothesized that urinary claudin1-positive cells would be a good indicator of crescent formation. As the major kidney infiltrating cells, we used CD68 as a marker for macrophages (MΦ) and neutrophil elastase (NE) as a marker for neutrophils. We further examined the expression levels of the mRNAs for these markers by real-time reverse transcription polymerase chain reaction (RT-PCR). We also examined the levels of mRNAs associated with the M1 (iNOS, IL-6) and M2 (CD163, CD204, CD206, IL-10) MΦ phenotypes. These evaluated markers were compared with clinical and renal histological findings for the assessment of renal diseases.

## 2. Results

### 2.1. Verification of Cytokeratin, Synaptopodin, and Claudin1 as Markers of Renal Parenchymal Cells

To confirm that the selected markers can be detected on renal parenchymal cells, we performed double IF staining for claudin1 and cytokeratin (Figure 1A–C), for claudin1 and synaptopodin (Figure 1D–F), and for CD68 and NE on the kidney tissues of patients with minor glomerular abnormalities (MGA) and patients with ANCA-associated vasculitis (AAV).

Basically, the specific positive immunostaining results for each cell marker were histologically seen on cells corresponding to each cell marker: claudin1 on PECs, cytokeratin on tubular epithelial cells, synaptopodin on podocytes, CD68 on infiltrating MΦ, and NE on infiltrating neutrophils. Thus, the reliability of these antibodies in using them as markers for their respective cells was confirmed. The claudin1-positive cells in the kidneys of MGA patients were observed as a layer on Bowman’s capsule, whereas those of AAV patients were observed as diffuse clusters in glomerular crescents. As we observed both claudin1-positive cells and cytokeratin-positive cells in glomerular crescents, we further evaluated the renal biopsy specimens of AAV patients stained for claudin1 and cytokeratin under confocal microscopy (Zeiss LSM510; Carl Zeiss, Göttingen, Germany). In most of the glomeruli, the predominant cell type in crescents was claudin1 single-positive (Figure 1G–I), and the second-most predominant was cytokeratin single-positive. Cells double-positive for claudin1 and cytokeratin could be found but were not predominant in most of the glomeruli. Claudin1 was strongly positive on PECs but was also weakly positive on portions of the tubules. Double staining of the urinary sediment cells for claudin1 and cytokeratin showed that cells double-positive for these markers were rarely found in any of the specimens.

### 2.2. Claudin1-Positive and CD68-Positive Cells in Urinary Sediments

IF staining of the urinary sediments from patients with various kidney diseases revealed that the number of claudin1-positive cells was significantly larger in AAV than in IgA nephropathy (IgAN), Henoch–Schönlein purpura nephritis (HSPN), and healthy volunteers (*p* < 0.05, Dunnett’s test) (Figure 2), but the number of CD68-positive cells did not differ significantly between AAV and other glomerular diseases or healthy volunteers. The number of NE-positive cells was significantly larger in AAV than in minimal change nephrotic syndrome (MCNS), focal segmental glomerulosclerosis (FSGS), membranous nephropathy (MN), IgAN, diabetic nephropathy (DMN), and healthy volunteers (*p* < 0.05, Dunnett’s test). We also IF-stained the urinary sediments of 100 patients with various renal diseases for cytokeratin and synaptopodin but did not find any significant between-disease differences.

We then evaluated the staining results with regard to their relation to the presence or absence of glomerular crescents and to the degree of glomerulosclerosis, regardless of the histological diagnosis. Representative photomicrographs of IF staining for claudin1 and CD68 are shown in Figure 3A,B. As shown in Figure 3C,D, the urinary claudin1-positive and CD68-positive cell counts were significantly larger in patients with glomerular crescents than in patients without crescents (*p* = 0.005 and *p* < 0.001, respectively). We also evaluated the data with regard to the state of the glomerular crescents (cellular crescents, fibrocellular crescents, fibrous crescents, and no crescents) and to the degree of crescent formation. As shown in Figure 3E,F, the urinary claudin1-positive and CD68-positive cell counts were significantly higher only in patients with cellular crescents than in patients without crescents (*p* = 0.003 and *p* < 0.001, respectively). The urinary claudin1-positive and CD68-positive cell counts did not differ significantly between patients without crescents and patients with either fibrocellular or fibrous crescents.

Claudin1- and CD68-positive cell counts in urinary sediment showed weak but significant positive correlations with the degree of crescent formation (*r* = 0.163, *p* = 0.024, and *r* = 0.149, *p* = 0.041, respectively). On the other hand, neither the urinary claudin1-positive nor CD68-positive cell count differed significantly between patients with low rates of global sclerosis and patients with high rates of global sclerosis (Figure 4). The urinary NE-positive cell count was also significantly larger in patients with glomerular crescents than in patients without crescents (*p* = 0.001), especially in patients with cellular crescents (*p* < 0.001), and it showed significant positive correlations with the degree of crescent formation (*r* = 0.505, *p* < 0.001). However, it did not differ significantly depending on the global sclerosis rate.

There was a significant positive trend between the urinary claudin1-positive cell count and the urinary protein level (*p* < 0.001), but the correlation between the urinary CD68-positive cell count and urinary protein level did not reach the level of statistical significance (*p* = 0.078) (Figure 5A,B). No significant correlation was found between the urinary NE-positive cell count and the urinary protein level (*p* = 0.209). On the other hand, the urinary claudin1-positive (Figure 5C), CD68-positive (Figure 5D), and NE-positive cell counts showed a significant positive correlation with the level of urinary RBC count (*p* = 0.011, *p* < 0.001, and *p* < 0.001, respectively).

We used logistic regression to evaluate this stain’s predictive value with regard to the formation of glomerular crescents. Logistic regression analysis of urinary claudin1- (Figure 6A), CD68- (Figure 6B), and NE-positive cell counts for the presence of glomerular crescents yielded receiver operating characteristic (ROC) curves with area under the curve (AUC) values of 0.603, 0.695, and 0.660, respectively, all of which were statistically significant (*p* = 0.005, *p* < 0.001, and *p* < 0.001, respectively). 

These AUC values, however, were found to be smaller than that of the ROC curve of urinary RBC count for the presence of glomerular crescents (0.699, *p* < 0.001) (Figure 6C). Therefore, we performed logistic regression analysis by using the composite scores of urinary RBC- and claudin1- (Figure 6D), urinary RBC- and CD68- (Figure 6E), and urinary RBC- and NE-positive cell counts for the presence of glomerular crescents, which yielded ROC curves with AUC values of 0.731, 0.753, and 0.730, respectively, all of which were significant (*p* = 0.02, *p* = 0.006, and *p* = 0.003, respectively) and were larger than the AUC values of ROC curves with each of the single markers (RBC counts, claudin1-, CD68-, or NE-positive cell counts).

### 2.3. Claudin1 and CD68 mRNA Levels in Urinary Sediment

The relative level of CD68 mRNA in urinary sediments was significantly higher in patients with glomerular crescents than in patients without crescents (*p =* 0.009) (Figure 7A), but the difference in the relative level of claudin1 mRNA between patients with glomerular crescents and patients without crescents was not statistically significant (*p* = 0.09) (Figure 7B). Logistic regression analysis of CD68 mRNA levels for the presence of glomerular crescents yielded the ROC curve with AUC value of 0.648, which was significant (*p* = 0.006) (Figure 7C), while that of claudin1 mRNA for the presence of glomerular crescents yielded the ROC curves with AUC value of 0.507, which was not significant (*p* = 0.241) (Figure 7D). The relative levels of claudin1 and CD68 mRNA in urinary sediments did not differ between patients with different rates of global sclerosis. NE mRNA was not detected in any specimens.

The relative levels of mRNA for the M2 MΦ markers (CD163, CD204, CD206, and IL-10) in urinary sediments were significantly higher in patients with crescents than in patients without crescents (Figure 8A–D). The relative levels of mRNA for M1 markers (iNOS, IL-6), on the other hand, were similar in patients with crescents and patients without crescents (Figure 8E,F). None of the MΦ marker mRNA levels showed any significant difference in relation to the global sclerosis rate. 

Logistic regression analysis of M2 marker (CD163, CD204, CD206, IL-10) mRNA levels in urinary sediments for the presence of glomerular crescents yielded ROC curves with AUC values of 0.667, 0.650, 0.569, and 0.614, respectively, all of which were significant (*p* < 0.001, *p* < 0.001, *p* < 0.001, and *p* = 0.006, respectively).

## 3. Discussion

The relationship between urinary podocytes and disease activity, both in patients with renal diseases and in experimental models, has been widely reported [6,7,8,9,10]. Few reports, however, have dealt with other cellular components of urine [11,12,13], and none have dealt with the claudin1-positive cells in urinary sediments.

Claudins are a family of tight-junction-forming proteins that have four transmembrane domains [15]. Kiuchi-Saishin et al. [14] examined the localization of claudin proteins in mouse kidneys and found claudin1 to be expressed specifically on glomerular PECs. Ohse et al. [16] proposed that PEC tight junctions serve as a barrier keeping the protein filtered by glomerular tufts out of the interstitium. When this barrier is disturbed by PEC injury, its permeability increases, possibly resulting in interstitial inflammation subsequent to periglomerular leakage of filtered protein. PECs are also involved in the formation of crescents [17,18,19], so claudin1-positive cells are expected to be excreted in the urine of patients with glomerular crescents.

The site of claudin1 expression may differ between human and mouse kidneys; although claudin1 expression in the mouse kidney is reportedly restricted to the glomerular PECs, claudin1 in the human kidney is reportedly also expressed in the distal tubules and collecting ducts [20]. Furthermore, inducible expression of claudin1 on podocytes has recently been reported in injured glomeruli of both mouse and human kidneys [21,22]. In the present study, we also found claudin1 staining not only in PECs but also in parts of tubules. The staining intensity in those tubules was weak, however, compared to that in the PECs. Small parts of crescent-forming cells were found to be double-positive for claudin1 and cytokeratin in the renal biopsy tissues of AAV patients. Double staining the urinary sediment cells for claudin1 and cytokeratin revealed that, in most cases, double-positive cells were quite rare. We therefore think that most of the claudin1-positive cells we found in urinary sediments were PECs, and some phenotypes changed podocytes.

We first assessed the relations of various renal diseases to claudin1-, CD68-, and NE-positive cell counts and found significant differences between only some of the diseases. Thus, we found it impossible to diagnose various renal diseases only by urinalysis, which means that a renal biopsy is still indispensable for making the final diagnosis. Renal biopsies, however, are performed not only to make final diagnoses but also to evaluate the state of renal disease (active or inactive, advanced or early). In deciding the use or discontinuation of immunosuppressants and glucocorticoids, the disease state is often more important than the final diagnosis. We therefore evaluated the urinary assays in relation to parameters reflecting the state of renal disease, such as the presence or absence of glomerular crescents, the state of glomerular crescents, and the degree of glomerulosclerosis.

As a result, we found that the numbers of claudin1-, CD68-, or NE-positive cells were significantly higher in patients with glomerular crescents, in particular with cellular crescents, than in patients without crescents. On the other hand, the numbers of these cells did not differ significantly between patients with different rates of global sclerosis. Furthermore, logistic analysis revealed that the numbers of claudin1-positive and CD68-positive urinary cells are significantly related to the likelihood of glomerular crescents. Actually, the AUC values of these markers were found to be smaller than that of the ROC curve of urinary RBC count for the presence of glomerular crescents. However, the AUC values of logistic regression of the composite scores using RBC- and claudin1-, RBC- and CD68-, and RBC- and NE-positive cell counts for the presence of glomerular crescents were larger than the AUC value of logistic regression of RBC count, which supports the additional value of these urinary sediment analyses for assessment of the state of glomerular diseases. The fact that the highest AUC value was found in the ROC curve of urinary RBC count for the presence of glomerular crescents suggests the clinical importance of urinary RBC in the assessment of glomerular diseases and somewhat supports the recent reports on the significance of hematuria in IgAN [23], AAV [24], and CKD [25].

The urinary protein levels in this study were significantly related only to the claudin1-positive cell count. That is, neither the CD68-positive cell count nor the NE-positive cell count were significantly related to the urinary protein level. Unlike neutrophils and MΦs, both of which are suspected to increase with the degree of intraglomerular inflammation, claudin1-positive cells may reflect the PEC damage or phenotype-changed podocytes associated with the overloaded protein due to the damage to the glomerular filtration barrier. 

As for the real-time RT-PCR, we at first used the GAPDH (primer/probe set was purchased from Applied Biosystems, Foster City, CA, USA; cat. No. Hs99999905_m1) as a reference gene. However, we could not find any significant difference in any of the mRNA levels between patients with crescents and those without crescents when we used GAPDH mRNA as the reference control. We suspected that urinary sediments may contain various cells that are not of kidney parenchymal origin, such as the ureter and bladder epithelium, and normalization using commonly-used internal controls such as GAPDH may mask positive results. Then, we used aquaporin2 (AQP2: primer/probe set was purchased from Applied Biosystems; cat. No. Hs00166640_m1) as the reference mRNA. AQP2 mRNA was reported to be a robustly expressed kidney-specific gene to represent kidney RNA from the non-glomerular compartment, and the rate of decay by time and temperature was similar for all mRNA [8]. The results from using AQP2 as the reference gene were better: The relative mRNA levels of claudin1 and CD68 with reference to AQP2 mRNA were similar to the results with the immunofluorescent staining for these cells. However, the limitation of this assay was that in about half of patients, AQP2 mRNA could not be detected probably due to the paucity of total RNA in urinary sediments, and we could not evaluate the mRNA assay in about half of patients. Therefore, we finally stopped using any reference control mRNA. We based the assessment of the mRNA expression level on the absolute Ct value of real-time PCR using total RNA isolated from the same volume (corresponding to 142 µL of original urine) of urine samples. The similarity of the claudin1- and CD68-mRNA relative-level results to the IF-staining results for the corresponding cells is evidence validating the present system of mRNA evaluation.

The difference in the relative levels of claudin1 mRNA between patients with glomerular crescents and patients without crescents did not reach statistical significance (*p* = 0.09). As described above, urinary claudin1 mRNA may also originate from the tubules [20]. PCR is a method of high sensitivity, so it might detect a small quantity of claudin1 mRNA from cells other than PECs and thereby overestimate the claudin1 mRNA levels of PECs. 

Regarding the urinary MΦs, we found a wide variation in cell size by IF staining for CD68 (Figure 9). A similar observation was reported by Oda et al. [26], who found characteristically large cells with macrophagic phenotypes (called intratubular giant MΦs) in the renal tubules and in the urine of patients with progressive glomerulonephritis. Various MΦ phenotypes related to function have been reported recently. Pro-inflammatory M1 MΦs exacerbate renal cell damage, while anti-inflammatory M2 MΦs promote epithelial and vascular repair [27,28]. We therefore evaluated the phenotypes of the MΦs in urinary sediments by RT-PCR. While the relative levels of mRNA for M2 MΦ markers were significantly higher in patients with glomerular crescents than in patients without crescents, the relative levels of mRNA for M1 MΦ markers did not differ between patients with and without crescents. Urinary M2 MΦs thus seem to play an important role in patients with glomerular crescents.

Even though we used four separate primer probe sets—human neutrophil elastase; human CD66b; human CD16a; and human myeloperoxidase—we were unable to detect any mRNA for neutrophil markers in total RNA isolated from urinary sediments. All of these neutrophil-related mRNAs could be detected in total RNA isolated from blood, however, which made us suspect that neutrophils in urine are terminally differentiated cells unable to synthesize new proteins.

Although the results of some markers for predicting disease state were statistically significant, the predictive value (AUC of ROC curves) of them was modest. This may be related to the limitations of this study. Given the sample collection and evaluation methods of this study, differences in the hydration status of each patient would affect the variability of urine samples and the strictness of the results. Furthermore, medications were different for each patient, and it is possible that these limitation factors led to the low AUC values. Other urinary clinical parameters, such as N-acetyl-β-D-glucosaminidase or β2-microglobulin, were not examined in this study, but the possibility of improving AUC values through combined evaluation with these markers should be evaluated in future studies.

In summary, renal biopsy cannot be carried out repeatedly and may not always be an optimal approach for the detection of glomerular crescents. We found the existence of glomerular crescents to be predicted using urinary sediments by IF staining for claudin1 and CD68 and by quantification of the levels of mRNA for CD68 or M2 MΦ markers. We, therefore, think that such risk-free examinations evaluating crescent formation would be very helpful in the management of renal diseases (such as deciding the usage or discontinuation of immunosuppressants and glucocorticoids), especially in children and the elderly, for whom a renal biopsy is not an easily endured procedure.

## 4. Materials and Methods

### 4.1. Patients

We collected morning urine specimens from 206 patients who had been hospitalized at the National Defense Medical College for a kidney biopsy between 2008 and 2011. The 6 patients whose biopsy specimen yielded fewer than 5 glomeruli were excluded from this study, as were the 8 patients with non-glomerular disease (6 with tubulointerstitial nephritis and 2 with nephrosclerosis). We finally assessed 192 urine samples from patients with various glomerular diseases. The characteristics of the patients enrolled in this study are listed in Table 1. 

Their diagnoses were based on clinical symptoms and laboratory data, as well as the histological features of renal biopsy tissues. The diagnoses of the patients are shown in Table 2. 

As the normal control, we also assessed 8 urine samples from healthy volunteers. This study was approved by the Ethical Committee of the National Defense Medical College on 1 September 2011 (No. 00934), and written informed consent was obtained from each patient in accordance with the principles of the Declaration of Helsinki.

### 4.2. Collection and Treatment of Urine Specimens

Forty milliliters of morning urine were collected from each patient, 30 mL of which was used to make glass slides. The urinary sediments in the 30 mL were obtained by centrifugation at 1500 rpm for 7 min and were washed in PBS. After the PBS was removed, the sediments were resuspended in 600 μL of 0.2% BSA-PBS. Then, the sediments in 100 μL of each resuspension were deposited on a glass slide by using a Cytospin 4 (Thermo Fisher Scientific, Pittsburgh, PA, USA) (5 min at 700 rpm). The slides were stored at −80 °C until use.

The other 10 mL of each sample were used for the isolation of total RNA. The sediments in them were similarly obtained by centrifugation, and 100 μL of the TRIzol Plus RNA isolation reagent (Thermo Fisher Scientific) was soon added to the sediments. The sediments suspended in TRIzol were stored at −80 °C until the process of RNA extraction.

### 4.3. IF Staining

Several selected renal biopsy tissues (from 2 patients with MGA and 8 patients with AAV) and all of the cytospin slides were IF-stained as described previously [29]. Details of the primary antibodies are listed in Table 3. After blocking with 10% normal goat serum, the slides were incubated with antibodies against claudin1, CD68, NE, synaptopodin, and pan-cytokeratin for 60 min at room temperature. After the slides were washed with PBS, secondary antibodies [goat anti-rabbit IgG AF594: red (Thermo Fisher Scientific) and goat anti-mouse IgG AF488: green (Thermo Fisher Scientific)] were added at a dilution of 1:150, and the slides were incubated for 60 min at room temperature. Counter-staining was performed using Hoechst 33342 (Merck, St. Louis, MO, USA). Five images (×100) of each slide were randomly captured using a digital camera. In each, the numbers of cells positive for various markers were counted, and the results for all 5 images were averaged.

### 4.4. RNA Extraction and Real-Time RT-PCR

Total RNA was extracted from the sediments stored suspended in TRIzol by using the TRIzol reagent following the manufacturer’s instructions (Thermo Fisher Scientific). Extracted RNA was dissolved in 50 μL of diethylpyrocarbonate-treated distilled water, and the RNA concentration and purity were determined by measuring absorbance at 260 nm and 280 nm using a NanoPhotometer (Implen, München, BY, Germany).

The RNA in 15 µL of each solution was reverse-transcribed using Transcriptor Universal cDNA Master (Roche, Basel, Switzerland) at a final volume of 20 μL, and the resulting complementary DNA (cDNA) in 1 μL was used as a template for real-time PCR with the primer/probe sets of TaqMan Gene Expression Assays (Applied Biosystems, Foster City, CA, USA). All of the primer/probe sets used in this study were purchased from Applied Biosystems: human claudin1 (cat. No. Hs00221623_m1), human CD68 (cat. No. Hs00154355_m1), human NE (cat. No. Hs00236952_m1), human CD66b (cat. No. Hs00266198_m1), human CD16a (cat. No. Hs01569121_m1), human myeloperoxidase (cat. No. Hs00924296_m1), human iNOS (cat. No. Hs01126940_gH), human IL-6 (cat. No. Hs00985639_m1), human CD163 (cat. No. Hs00174705_m1), human CD204 (cat. No. Hs00234007_m1), human CD206 (cat. No. Hs00267207_m1), and human IL10 (cat. No. Hs00961622_m1). Real-time PCR was performed using a StepOnePlus system (Applied Biosystems), and the comparative Ct (ΔCt) method described by the system’s manufacturer was used to determine relative mRNA values for each patient by expressing the absolute Ct values of real-time PCR for a patient as values relative to the absolute Ct values of one specific patient (one with IgAN).

### 4.5. Statistical Analysis

Data are expressed as mean ± standard error (SE). The significance of differences between the two groups was evaluated using an unpaired Student’s *t*-test, a one-way ANOVA followed by the Tukey–Kramer test for comparisons between four groups, and Dunnett’s test for comparisons between various renal diseases.

Logistic regression analysis was performed. ROC curves, the areas under them, and odds ratios were calculated. 

The correlations between the two consecutive variables were evaluated using the Pearson correlation coefficient, while those between consecutive variables and categorical variables were examined using the Jonckheere–Terpstra trend test.

Statistical analyses other than the Jonckheere–Terpstra trend test, which was performed with EZR software version 1.27 (Saitama Medical Center, Jichi Medical University, Saitama, Japan) [30], were performed with JMP ver. 10.0.2 software (SAS Institute Inc., Cary, NC, USA). *p* values < 0.05 were considered statistically significant.

## Figures and Tables

**Figure 1 ijms-25-00744-f001:**
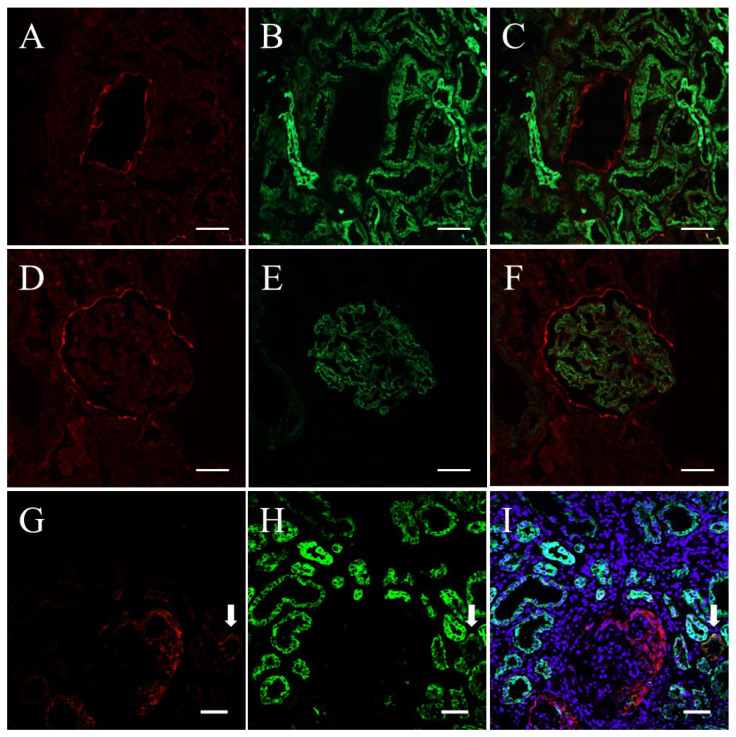
Representative photomicrographs of double staining for claudin1 and cytokeratin and for claudin1 and synaptopodin in kidney tissues of patients with minor glomerular abnormality (MGA) (**A**–**F**) and confocal microscopic images of double staining for claudin1 and cytokeratin in kidney tissues of a patient with ANCA-associated vasculitis (AAV) (**G**–**I**). (**A**): claudin1 labeled with Alexa Fluor 594, red; (**B**): cytokeratin labeled with Alexa Fluor 488, green. (**C**): A merged with B. (**D**): claudin1 labeled with Alexa Fluor 594, red. (**E**): synaptopodin labeled with Alexa Fluor 488, green. (**F**): D merged with E (scale bars = 50 µm). (**G**): claudin1 labeled with Alexa Fluor 594, red. (**H**): cytokeratin labeled with Alexa Fluor 488, green. (**I**): G merged with H and Hoechst 33342 nuclear staining, blue. (scale bars = 50 µm). In the kidneys of MGA patients, claudin1-positive cells were observed as a layer on Bowman’s capsule. Cells double-positive for claudin1 and cytokeratin were occasionally observed in small portions of renal tubules, but cells double-positive for claudin1 and synaptopodin were rarely found in any of the specimens (**A**–**F**). Under confocal microscopy, claudin1-positive cells were the major component of crescent-forming cells in most glomeruli, but some of the cytokeratin-positive cells were also observed in part of a crescent. Tubular epithelial cells that were double-positive for claudin1 and cytokeratin were occasionally observed (**G**–**I**: arrows).

**Figure 2 ijms-25-00744-f002:**
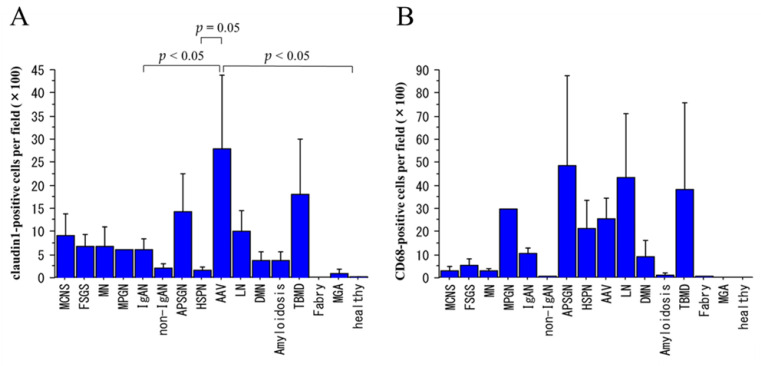
Urinary claudin1- and CD68-positive cell counts in various renal diseases and in healthy volunteers. MCNS: minimal change nephrotic syndrome; FSGS: focal segmental glomerulosclerosis; MN: membranous nephropathy; MPGN: membranoproliferative glomerulonephritis; IgAN: IgA nephropathy; APSGN: acute poststreptococcal glomerulonephritis; HSPN: Henoch–Schönlein purpura nephritis; AAV: ANCA-associated vasculitis; LN: lupus nephritis; DMN: diabetic nephropathy; TBMD: thin basement membrane disease; MGA: minor glomerular abnormality. The number of cases for each disease is listed in Table 2, and the number of healthy volunteers is 8. Data are shown as mean ± SE. (**A**): The claudin1-positive cell count was significantly higher in patients with AAV than in patients with IgAN, HSPN, and healthy volunteers (*p* < 0.05, Dunnett’s test). (**B**): The CD68-positive cell count did not differ significantly between AAV and other diseases (Dunnett’s test).

**Figure 3 ijms-25-00744-f003:**
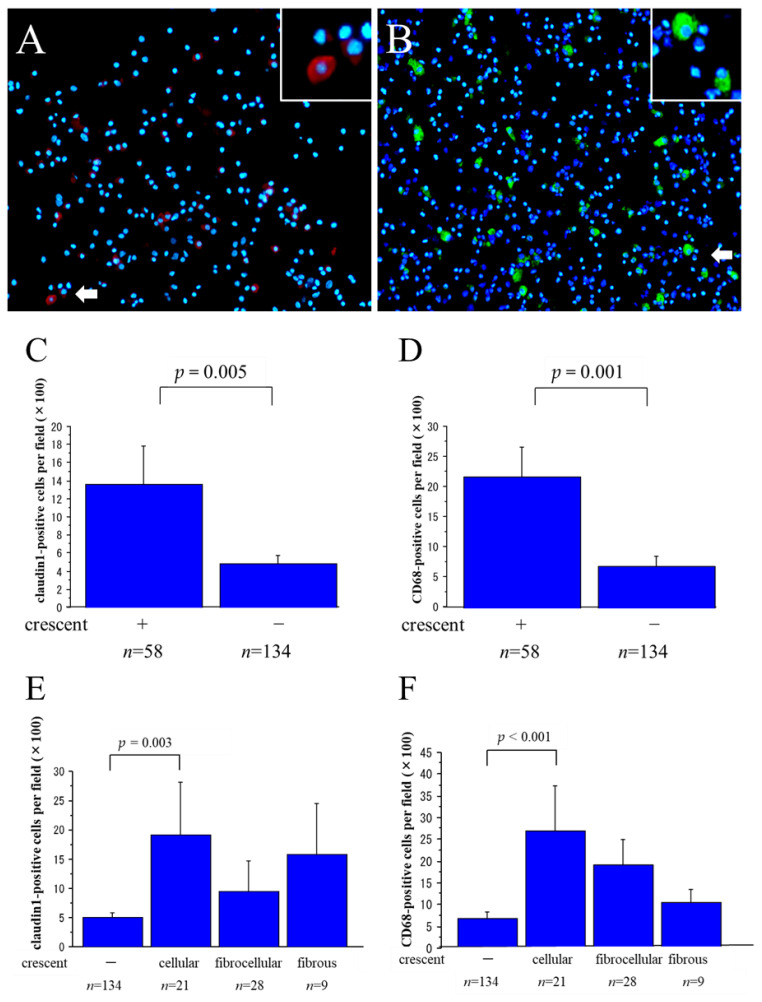
Immunofluorescence (IF) staining for claudin1 and CD68 in the cytospin slides of urinary sediments. (**A**,**B**): Representative photomicrographs of IF staining for claudin1 labeled with Alexa Fluor 594, red (**A**) and CD68 labeled with Alexa Fluor 488, green (**B**) and Hoechst 33342 nuclear staining, blue with insets of higher-magnification views of areas indicated by white arrows in the urinary sediments of a patient with ANCA-associated vasculitis. (scale bars = 50 µm). Large numbers of claudin1-positive cells and CD68-positive cells were observed in patients with glomerular crescents. The claudin1-positive cell count (**C**) and CD68-positive cell count (**D**) were significantly higher in patients with glomerular crescents than in patients without crescents (*p* = 0.005 and *p* < 0.001, respectively, Student’s *t*-test). In particular, the claudin1-positive cell count (**E**) and CD68-positive cell count (**F**) were significantly higher in patients with cellular crescents than in patients without crescents (*p* = 0.003 and *p* < 0.001, respectively, Tukey–Kramer test); however, the differences between patients without crescents and patients with fibrocellular or fibrous crescents were not significant.

**Figure 4 ijms-25-00744-f004:**
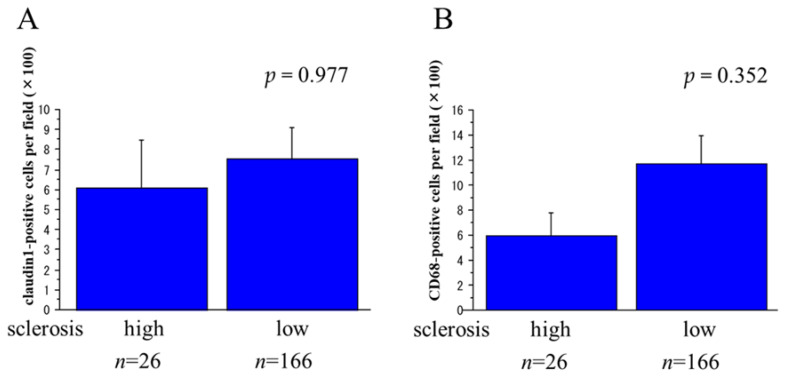
Relations between the urinary claudin1- and CD68-positive cell counts and the degree of glomerulosclerosis. Neither the claudin1-positive cell count (**A**) nor the CD68-positive cell count (**B**) differed significantly in relation to the global sclerosis rate (high = global sclerosis found in more than 30% of glomeruli; low = global sclerosis found in less than 30% of glomeruli).

**Figure 5 ijms-25-00744-f005:**
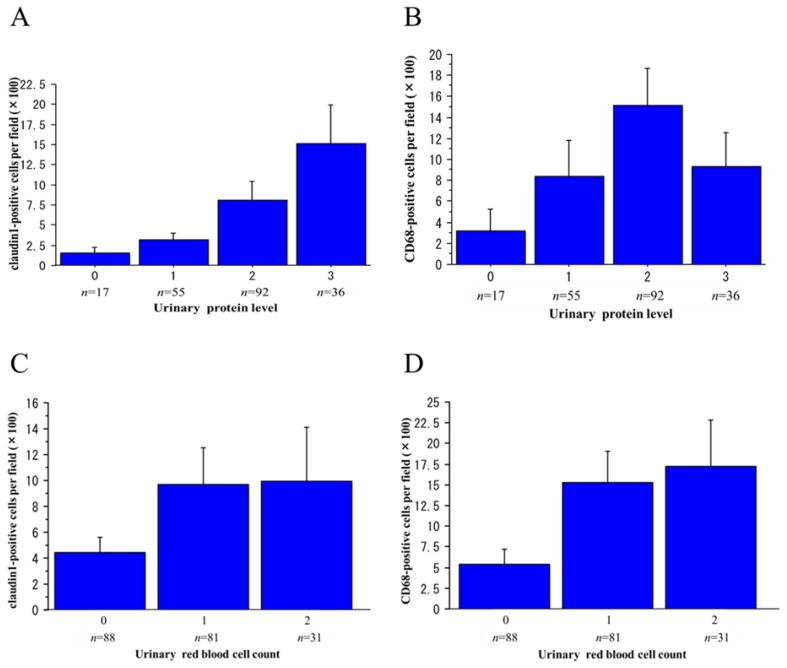
Relations between urinary protein and urinary red blood cell (RBC) levels and the claudin1- and CD68-positive cell counts in urinary sediments. Urinary protein level (g/gCr); 0: <0.15, 1: ≧0.15, <0.5, 2: ≧0.5, <3.5, 3: ≧3.5. Urinary RBC count (HPF): 0: 0–4, 1: 5–29, 2: ≧30. There was a significant positive trend between claudin1-positive cell count and urinary protein level (*p* < 0.001, Jonckheere–Terpstra test) (**A**), but not between CD68-positive cell count and urinary protein level (*p* = 0.078, Jonckheere–Terpstra test) (**B**). On the other hand, both the claudin1- and CD68-positive cell counts showed significant positive trends with the level of urinary RBC count (*p* = 0.011 and *p* < 0.001, respectively, Jonckheere–Terpstra test) (**C**,**D**).

**Figure 6 ijms-25-00744-f006:**
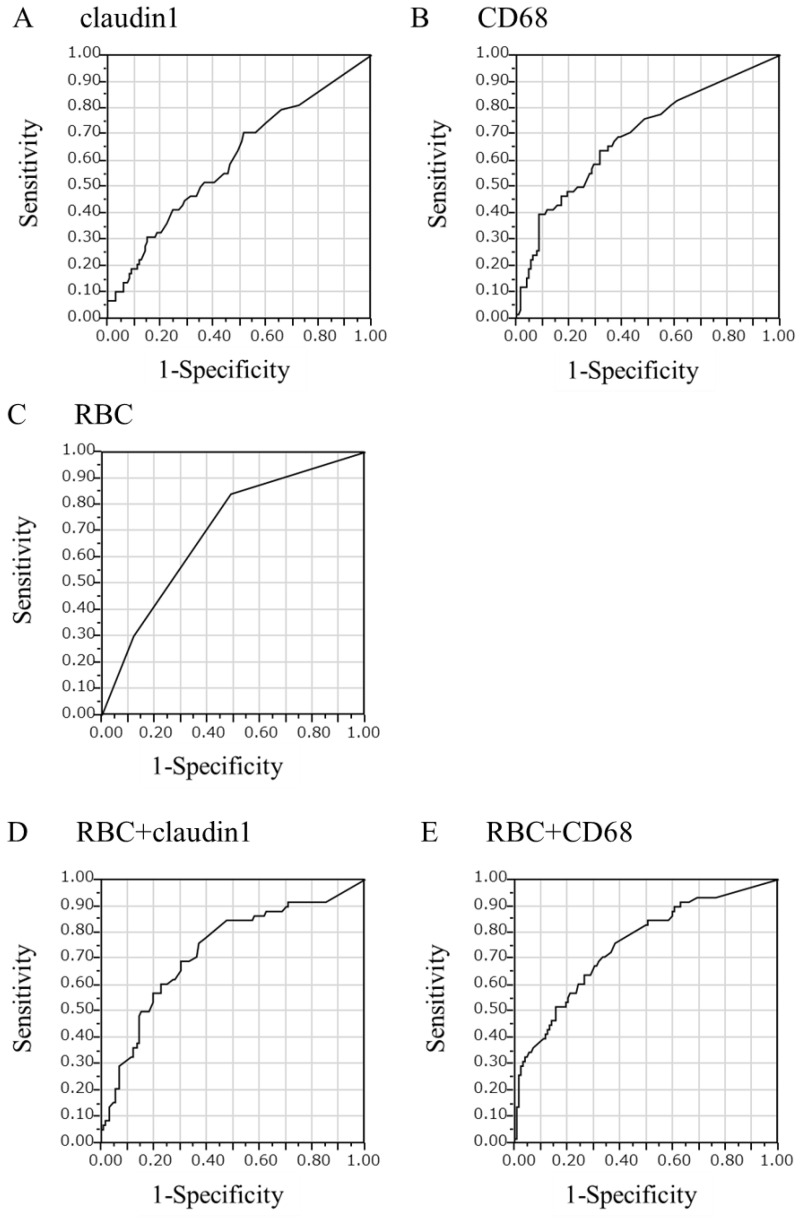
Receiver operating characteristic (ROC) curves for the prediction of glomerular crescents. Logistic regression analysis of urinary claudin1-positive cell count (**A**), CD68-positive cell count (**B**), and red blood cell (RBC) count (**C**) for the presence of glomerular crescents yielded ROC curves with area under the curve (AUC) values of 0.603, 0.695, and 0.699, respectively, which were statistically significant (*p* = 0.005, *p* < 0.001, and *p* < 0.001, respectively). Logistic regression analysis of composite scores of RBC- and claudin1-positive cell counts (**D**) and RBC- and CD68-positive cell counts (**E**) for the presence of glomerular crescents yielded ROC curves with AUC values of 0.731 and 0.753, respectively, all of which were statistically significant (*p* = 0.02 and *p* = 0.006, respectively).

**Figure 7 ijms-25-00744-f007:**
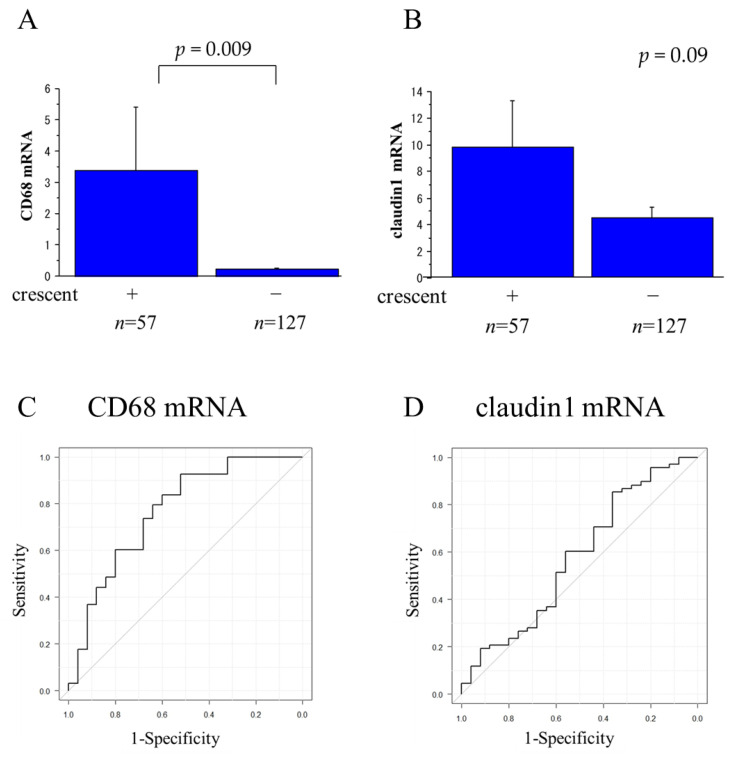
Relations between the CD68 and claudin1 mRNA levels and the presence of glomerular crescents, and ROC curves for the prediction of glomerular crescents by these markers. The CD68 mRNA level was significantly higher in patients with glomerular crescents than in patients without crescents (*p* = 0.009) (**A**), but the difference between them was not significant (*p* = 0.09) for the claudin1 mRNA level (**B**). Logistic regression analysis of CD68 mRNA level in urinary sediments for the presence of glomerular crescents yielded ROC curves with an AUC value of 0.648, which was statistically significant (*p* = 0.006) (**C**), but for claudin1 mRNA, the AUC value was 0.507 which was not significant (*p* = 0.241) (**D**).

**Figure 8 ijms-25-00744-f008:**
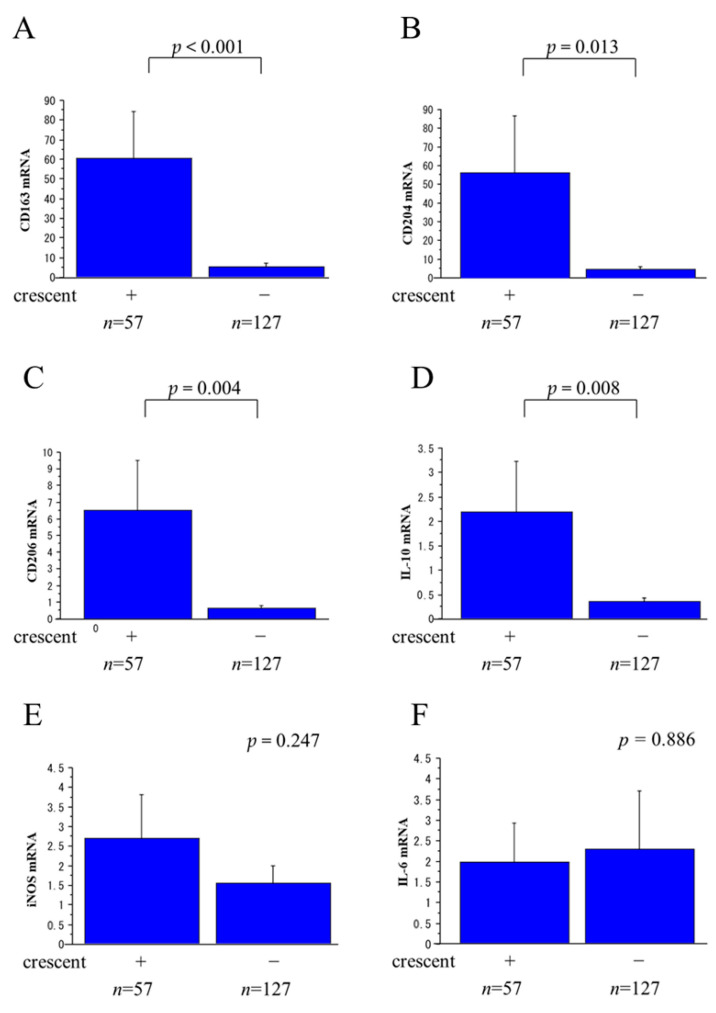
Relations between crescents and levels of mRNA for M1/M2 macrophage (MΦ) markers in urinary sediments. M1 MΦ markers: iNOS, IL-6. M2 MΦ markers: CD163, CD204, CD206, IL-10. The relative levels of mRNA for CD163 (**A**), CD204 (**B**), CD206 (**C**), and IL-10 (**D**) were significantly higher in patients with crescents than patients without crescents. On the other hand, the relative levels of mRNA for iNOS (**E**) and IL-6 (**F**) did not differ between patients with crescents and patients without crescents.

**Figure 9 ijms-25-00744-f009:**
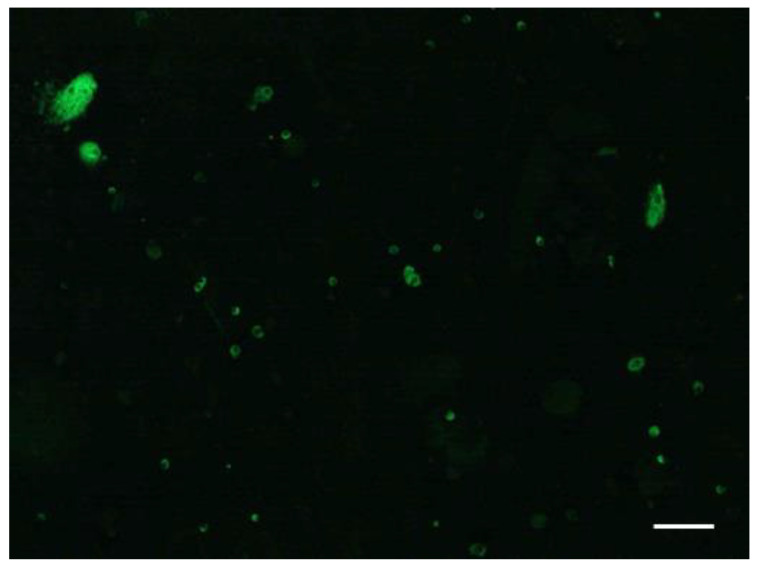
Representative IF staining for CD68 in urinary sediments of AAV patients. In IF staining of urinary sediments, there was variation in the size of CD68-positive cells (scale bar = 50 µm).

**Table 1 ijms-25-00744-t001:** Characteristics of the patients enrolled in this study.

Sex (male/female)	100/92
Age (years)	44 ± 18
Serum creatinine (mg/dL)	1.13 ± 1.25
Urinary red blood cell (URBC) count (0/1/2)	88/81/31
Urinary protein (g/gCr)	2.14 ± 3.32
Serum albumin (g/dL)	3.43 ± 0.99

Data are presented as the mean ± standard error or number. The URBC count was semiquantitatively graded as follows: 0, URBC count: 0–4; 1, URBC count: 5–29; 2, URBC count: ≧30.

**Table 2 ijms-25-00744-t002:** The diagnoses of the patients enrolled in this study.

	Number
Minimal change nephrotic syndrome (MCNS)	15 (1)
Focal segmental glomerulosclerosis (FSGS)	22 (1)
Membranous nephropathy (MN)	19 (3)
Membranoproliferative glomerulonephritis (MPGN)	1
IgA nephropathy (IgAN)	73 (32)
Mesangial proliferative non-IgA nephropathy (non-IgAN)	5
Acute poststreptococcal glomerulonephritis (APSGN)	4 (1)
Henoch–Schönlein purpura nephritis (HSPN)	10 (6)
ANCA-associated vasculitis (AAV)	11 (11)
Lupus nephritis (LN)	9 (2)
Diabetic nephropathy (DMN)	8
Amyloidosis	5 (1)
Thin basement membrane disease (TBMD)	3
Fabry disease	1
Minor glomerular abnormality (MGA)	6
Total	192 (58)

The number of patients with glomerular crescents are shown in parentheses.

**Table 3 ijms-25-00744-t003:** Primary antibodies used in this study.

Antibody Type	Clone	Specificity	Supplier	Dilution
rabbit anti-claudin1	-	parietal epithelial cells	Bioworld Technology, Minneapolis, MN, USA	1:100
mouse anti-CD68	EBM11	macrophages	Dako, Glostrup, Denmark	1:100
rabbit anti- neutrophil elastase	-	neutrophils	Calbiochem, La Jolla, CA, USA	1:500
mouse anti-synaptopodin	G1D4	podocytes	Progen, Heidelberg, Germany	1:1
mouse anti-cytokeratin	KL-1	tubular epithelial cells	IMMUNOTECH, Monrovia, CA, USA	1:100

## Data Availability

The data presented in this study are available upon request from the corresponding author. The data are not publicly available due to ethical and privacy limitations.

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
