# Peer review of "Evaluating the State of Glomerular Disease by Analyzing Urinary Sediments: mRNA Levels and Immunofluorescence Staining for Various Markers"

_ijms, 2024, doi:10.3390/ijms25020744_

Round 1

Reviewer 1 Report

Comments and Suggestions for Authors

This manuscript presents a novel approach to evaluate glomerular diseases by analyzing urinary sediments for claudin1 and CD68, both at the protein and mRNA levels. The study is grounded in a robust experimental design involving 192 patients, providing a substantial dataset for analysis. Overall, this manuscript is well written and methods are straightforward and clearly described. For this reason, I only have some minor suggestions to improve the manuscript:

1.      The authors mention using total RNA from sediments without a reference control gene. While the rationale is explained, it may be worthwhile to explore if there is a more suitable housekeeping gene that could normalize the data more accurately.

2.      Though the results are statistically significant, the predictive value (AUC of ROC curves) is modest. Discuss whether combining these markers with other clinical parameters could improve diagnostic accuracy.

3.      Please discuss the potential variability in urine samples due to factors such as hydration status or concurrent medications.

Check and make sure that each staining picture has appropriate scale bar labeled.

Reviewer 2 Report

Comments and Suggestions for Authors Dear authors,

I thank you for this elegant research. I found it interesting to find new ways to diagnose kidney disease without doing biopsy.

Main remark:

1/ Nevertheless, you and also others researchers have to be sure of your antibodies and primers used. Looking the antibodies you used I did not find where in the article you demonstrated that the staining is correct with positive controls. Thus, to my point of view you jump to fast to the conclusion in the sentence: "The specific and successful results obtained with these stainings confirmed that antibodies for these markers can be used to examine urinary sediments for the presence of cells with these markers". 

2/ Looking the primers used for the RT-qPCR, did you find the primers in specific publications or you designed yourself? How did you test these primers?

Second line remark:

1/ I have a doubt looking the number of healthy volunteers. Is the number sufficient with regard to inter-individual variation?
2/ In materials and methods sections, can you please indicate which T-test student you used?
3/ Please, indicate results in dot-plots?

Reviewer 3 Report

Comments and Suggestions for Authors

Yamamoto et. al. investigate the association between severe glomerular injury (glomerular crescents) and various markers, including cellular and inflammatory markers.

Abstract:

The abstract is written very chaotically. It start with a general statement that urine analysis may be better that kidney biopsy for diagnostics. Then the abstract continues with the method followed by the introduction. It seems to a reader that the markers are randomly chosen without any information on rationale. Result part of the abstract tries to describe the observed differences in markers, but it is impossible to follow whether authors mean histology or urine analysis. The abstract must be completely re-written. 

Line 34: 'This results' is not correct. There are multiple typos in the text. 

Introduction: it is only in this section (first paragraph) where it is clear what authors intended to investigate. This has to be pointed out in the abstract.

The introduction lacks information about kidney injury (pathophysiology). It will be helpful to point this out.

It will also be helpful to point out what the markers the authors assessed mean for the disease state. 

Line 104: what does 'successful results' mean?

Where the markers obtained with immunostaining form the patients with different kidney disease quantified? 

What is number of samples for each kidney disease (Figure 2)?

Line 168: what does 'weak but significant' mean, according to r or P values?

Line 178: What does ' significant positive trend' mean?

To better judge about the significance of these findings, the authors should present the findings in histology versus urine analysis. 

The authors talk about claudins in the discussion the first time, without proper introduction in the earlier sections of the manuscript.

Comments on the Quality of English Language

English quality is very poor.

Reviewer 4 Report

Comments and Suggestions for Authors

Summary statement

Yamamoto et al examined the feasibility of using urinary biomarkers such as claudin1, CD68, and other markers of inflammation to evaluate the extent of glomerular disease and predict the development of glomerular crescents. The investigators assess the levels of these biomarkers at both the gene transcript (via quantitative real-time RT-PCR) and protein (via immunohistochemistry) levels.  One highlighted utility of the study is the benefit of the noninvasive nature of urinalysis compared to renal biopsy.

Strengths of the study

Overall, the manuscript is well-written and cohesive.

The methods employed are clear and well-detailed.

The authors are forthcoming about several imitations of the study, including their issue with identifying a proper housekeeping gene for the real-time RT-PCR experiments.

Areas to be improved

Major points

The authors should consider reworking the title to reflect that several other markers besides claudin1 and CD68 were assessed by urinalysis.

Although described in the “Discussion” section, the rationale for choosing the assessed markers (i.e., claudin1, CD68, macrophage phenotype markers, etc.) should be briefly discussed in the “Introduction” section for the non-expert reader.

For Figure 2, the legend fonts should be enlarged for legibility. Also, statistical significance in Figure 2A (AAV group) is discussed in the figure legend but not annotated in the figure.

What is the authors’ reason for not presenting any of the data discussed in Section 2.3 of the “Results” section?  There are other supplemental data provided.

Minor points

None.

Round 2

Reviewer 3 Report

Comments and Suggestions for Authors

I have no further comments.